# Semi-Implicit Graph Variational Auto-Encoders

**Arman Hasanzadeh**[†*], **Ehsan Hajiramezanali**[†*]**, Nick Duffield**[†]**, Krishna Narayanan**[†]**,**
**Mingyuan Zhou**[‡]**, Xiaoning Qian**[†]

† Department of Electrical and Computer Engineering, Texas A&M University
{armanihm, ehsanr, duffieldng, krn, xqian}@tamu.edu
‡ McCombs School of Business, The University of Texas at Austin
mingyuan.zhou@mccombs.utexas.edu

## Abstract

Semi-implicit graph variational auto-encoder (SIG-VAE) is proposed to expand the flexibility of variational graph auto-encoders (VGAE) to model graph data. SIG-VAE employs a hierarchical variational framework to enable neighboring node sharing for better generative modeling of graph dependency structure, together with a Bernoulli-Poisson link decoder. Not only does this hierarchical construction provide a more flexible generative graph model to better capture real-world graph properties, but also does SIG-VAE naturally lead to semi-implicit hierarchical variational inference that allows faithful modeling of implicit posteriors of given graph data, which may exhibit heavy tails, multiple modes, skewness, and rich dependency structures. SIG-VAE integrates a carefully designed generative model, well suited to model real-world sparse graphs, and a sophisticated variational inference network, which propagates the graph structural information and distribution uncertainty to capture complex posteriors. SIG-VAE clearly outperforms a simple combination of VGAE with variational inference, including semi-implicit variational inference (SIVI) or normalizing flow (NF), which does not propagate uncertainty in its inference network, and provides more interpretable latent representations than VGAE does. Extensive experiments with a variety of graph data show that SIG-VAE significantly outperforms state-of-the-art methods on several different graph analytic tasks.

## 1 Introduction

Analyzing graph data is an important machine learning task with a wide variety of applications. Transportation networks, social networks, gene co-expression networks, and recommendation systems are a few example datasets that can be modeled as graphs, where each node represents an agent (e.g., road intersection, person, and gene) and the edges manifest the interactions between the agents. The main challenge for analyzing graph datasets for link prediction, clustering, or node classification, is how to deploy graph structural information in the model. Graph representation learning aims to summarize the graph structural information by a feature vector in a low-dimensional latent space, which can be used in downstream analytic tasks.

While the vast majority of existing methods assume that each node is embedded to a deterministic point in the latent space [5, 2, 25, 30, 14, 15, 10, 4], modeling uncertainty is of crucial importance in many applications, including physics and biology. For example, when link prediction in Knowledge Graphs is used for driving expensive pharmaceutical experiments, it would be beneficial to know what is the confidence level of a model in its prediction. To address this, variational graph auto-encoder (VGAE) [18] embeds each node to a random variable in the latent space. Despite its

---

popularity, 1) the Gaussian assumption imposed on the variational distribution restricts its variational inference flexibility when the true posterior distribution given a graph clearly violates the Gaussian assumption; 2) the adopted inner-product decoder restricts its generative model flexibility. While recent study tries to address the first problem by changing the prior distribution but does not show much practical success [11], the latter one is not well-studied yet to the best of our knowledge.

Inspired by recently developed semi-implicit variational inference (SIVI) [39] and normalizing flow (NF) [27, 17, 23], which offer the interesting combination of flexible posterior distribution and effective optimization, we propose a hierarchical variational graph framework for node embedding of graph structured data, notably increasing the expressiveness of the posterior distribution for each node in the latent space. SIVI enriches mean-field variational inference with a flexible (implicit) mixing distribution. NF transforms a simple Gaussian random variable through a sequence of invertible differentiable functions with tractable Jacobians. While NF restricts the mixing distribution in the hierarchy to have an explicit probability density function, SIVI does not impose such a constraint. Both SIVI and NF can model complex posterior distributions, which will help when the underlying true embedded node distribution exhibits heavy tails and/or multiple modes. We further argue that the graph structure cannot be fully exploited by the posterior distribution from the trivial combination of SIVI and/or NF with VGAE, if not integrating graph neighborhood information. On the other hand, it does not address the flexibility of the generative model as stated as the second VGAE problem.

To address the aforementioned issues, instead of explicitly choosing the posterior distribution family in previous works [18, 11], our hierarchical variational framework adopts a stochastic generative node embedding model that can learn implicit posteriors while maintaining simple optimization. Specifically, we innovate a semi-implicit hierarchical construction to model the posterior distribution to best fit both the graph topology and node attributes given graphs. With SIVI, even if the posterior is not tractable, its density can be evaluated with Monte Carlo estimation, enabling efficient model inference on top of highly enhanced model flexibility/expressive power. Our semi-implicit graph variational auto-encoder (SIG-VAE) can well model heavy tails, skewness, multimodality, and other characteristics that are exhibited by the posterior but failed to be captured by existing VGAEs. Furthermore, a Bernoulli-Poisson link function [41] is adopted in the decoder of SIG-VAE to increase the flexibility of the generative model and better capture graph properties of real-world networks that are often sparse. SIG-VAE facilitates end-to-end learning for various graph analytic tasks evaluated in our experiments. For link prediction, SIG-VAE consistently outperforms state-of-the-art methods by a large margin. It is also comparable with state-of-the-arts when modified to perform two additional tasks, node classification and graph clustering, even though node classification is more suitable to be solved by supervised learning methods. We further show that the new decoder is able to generate sparse random graphs whose statistics closely resemble those of real-world graph data. These results clearly demonstrate the great practical values of SIG-VAE. The implementation of our proposed model is accessible at `https://github.com/sigvae/SIGraphVAE`.

## 2 Background

**Variational graph auto-encoder (VGAE).** Many node embedding methods derive deterministic latent representations [14, 15, 10]. By expanding the variational auto-encoder (VAE) notion to graphs, Kipf and Welling [18] propose to solve the following problem by embedding the nodes to Gaussian random vectors in the latent space.

**Problem 1.** Given a graph $G = (\mathcal{V}, \mathcal{E})$ with the adjacency matrix $\mathbf{A}$ and $M$-dimensional node attributes $\mathbf{X} \in \mathbb{R}^{N \times M}$, find the probability distribution of the latent representation of nodes $\mathbf{Z} \in \mathbb{R}^{N \times \mathscr{L}}$, i.e., $p(\mathbf{Z} \,|\, \mathbf{X}, \mathbf{A})$.

Finding the true posterior, $p(\mathbf{Z} \,|\, \mathbf{X}, \mathbf{A})$, is often difficult and intractable. In Kipf and Welling [18], it is approximated by a Gaussian distribution, $q(\mathbf{Z} \,|\, \psi) = \prod_{i=1}^{N} q_i(\mathbf{z}_i \,|\, \psi_i)$ and $q_i(\mathbf{z}_i \,|\, \psi_i) = \mathcal{N}(\mathbf{z}_i \,|\, \psi_i)$ with $\psi_i = \{\boldsymbol{\mu}_i, \operatorname{diag}(\boldsymbol{\sigma}_i^2)\}$. Here, $\boldsymbol{\mu}_i$ and $\boldsymbol{\sigma}_i$ are $l$-dimensional mean and standard deviation vectors corresponding to node $i$, respectively. The parameters of $q(\mathbf{Z} \,|\, \psi)$, i.e., $\psi = \{\psi_i\}_{i=1}^{N}$, are modeled and learned using two graph convolutional neural networks (GCNs) [19]. More precisely, $\boldsymbol{\mu} = \mathbf{GCN}_{\boldsymbol{\mu}}(\mathbf{X}, \mathbf{A})$, $\log \boldsymbol{\sigma} = \mathbf{GCN}_{\boldsymbol{\sigma}}(\mathbf{X}, \mathbf{A})$ and $\boldsymbol{\mu}$ and $\boldsymbol{\sigma}$ are matrices of $\boldsymbol{\mu}_i$'s and $\boldsymbol{\sigma}_i$'s, respectively. Given $\mathbf{Z}$, the decoder in VGAE is a simple inner-product decoder as $p(A_{i,j} = 1 \,|\, \mathbf{z}_i, \mathbf{z}_j) = \operatorname{sigmoid}(\mathbf{z}_i \, \mathbf{z}_j^T)$.

The parameters of the model are found by optimizing the well known evidence lower bound (ELBO) [16, 7, 8, 33]: $\mathcal{L} = \mathbb{E}_{q(\mathbf{Z}\,|\,\psi)}[p(\mathbf{A}\,|\,\mathbf{Z})] - \mathbf{KL}[q(\mathbf{Z}\,|\,\psi)\,||\,p(\mathbf{Z})]$. Note that $q(\mathbf{Z}\,|\,\psi)$ here is equivalent to $q(\mathbf{Z}\,|\,\mathbf{X},\mathbf{A})$. Despite promising results shown by VGAE, a well-known issue in variational inference is underestimating the variance of the posterior. The reason behind this is the mismatch between the representation power of the variational family to which $q$ is restricted and the complexity of the true posterior, in addition to the use of $\mathbf{KL}$ divergence, which is asymmetric, to measure how different $q$ is from the true posterior.

**Semi-implicit variational inference (SIVI).** To well characterize the posterior while maintaining simple optimization, semi-implicit variational inference (SIVI) has been proposed by Yin and Zhou [39], which is also related to the hierarchical variational inference [26] and auxiliary deep generative models [21]; see Yin and Zhou [39] for more details about their connections and differences. It has been shown that SIVI can capture complex posterior distributions like multimodal or skewed distributions, which can not be captured by a vanilla VI due to its restricted exponential family assumption over both the prior and posterior in the latent space. SIVI assumes that $\psi$, the parameters of the posterior, are drawn from an implicit distribution rather than being analytic. This hierarchical construction enables flexible mixture modeling and allows to have more complex posteriors while maintaining simple optimization for model inference. More specifically, $\mathbf{Z} \sim q(\mathbf{Z}\,|\,\psi)$ and $\psi \sim q_\phi(\psi)$ with $\phi$ denoting the distribution parameters to be inferred. Marginalizing $\psi$ out leads to the random variables $\mathbf{Z}$ drawn from a distribution family $\mathcal{H}$ indexed by variational parameters $\phi$, expressed as

$$\mathcal{H} = \left\{ h_\phi(\mathbf{Z}) \ : \ h_\phi(\mathbf{Z}) = \int_\psi q(\mathbf{Z}\,|\,\psi) q_\phi(\psi) \, d\psi \right\}. \tag{1}$$

The importance of semi-implicit formulation is that while the original posterior $q(\mathbf{Z}\,|\,\psi)$ is explicit and analytic, the marginal distribution, $h_\phi(\mathbf{Z})$ is often implicit. Note that, if $q_\phi$ equals a delta function, then $h_\phi$ is an explicit distribution. Unlike regular variational inference that assumes independent latent dimensions, semi-implicit does not impose such a constraint. This enables the semi-implicit variational distributions to model very complex multivariate distributions.

Since the marginal probability density function $h_\phi(\mathbf{Z})$ is often intractable, SIVI derives a lower bound for ELBO, as follows, to optimize the variational parameters.

$$\begin{aligned}
\mathcal{L} = \mathbb{E}_{\mathbf{Z}\sim h_\phi(\mathbf{Z})} \left[ \log \frac{p(\mathbf{Y},\mathbf{Z})}{h_\phi(\mathbf{Z})} \right] &= -\mathbf{KL}(\mathbb{E}_{\psi\sim q_\phi(\psi)}[q(\mathbf{Z}\,|\,\psi)]\,||\,p(\mathbf{Z}\,|\,\mathbf{Y})) + \log p(\mathbf{Y}) \\
&\geq -\mathbb{E}_{\psi\sim q_\phi(\psi)}\mathbf{KL}(q(\mathbf{Z}\,|\,\psi)\,||\,p(\mathbf{Z}\,|\,\mathbf{Y})) + \log p(\mathbf{Y}) \\
&= \mathbb{E}_{\psi\sim q_\phi(\psi)} \left[ \mathbb{E}_{\mathbf{Z}\sim q(\mathbf{Z}\,|\,\psi)} \left[ \log \left( \frac{p(\mathbf{Y},\mathbf{Z})}{q(\mathbf{Z}\,|\,\psi)} \right) \right] \right] = \underline{\mathcal{L}}(q(\mathbf{Z}\,|\,\psi), q_\phi(\psi)),
\end{aligned} \tag{2}$$

where $\mathbf{Y}$ is the observations. The inequality $\mathbb{E}_\psi \mathbf{KL}(q(\mathbf{Z}\,|\,\psi)||p(\mathbf{Z})) \geq \mathbf{KL}(\mathbb{E}_\psi[q(\mathbf{Z}|\psi)]||p(\mathbf{Z}))$ has been used to derive $\underline{\mathcal{L}}$. Optimizing this lower bound, however, could drive the mixing distribution $q_\phi(\psi)$ towards a point mass density. To address the degeneracy issue, SIVI adds a nonnegative regularization term, leading to a surrogate ELBO that is asymptotically exact [39]. We will further discuss this in the supplementary material.

**Normalizing flow (NF).** NF [23] also enriches the posterior distribution families. Compared to SIVI, NF imposes explicit density functions for the mixing distributions in the hierarchy while SIVI only requires $q_\phi$ to be reparameterizable. This makes SIVI more flexible, especially when using it for graph analytics as explained in the next section, since the SIVI posterior can be generated by transforming random noise using any flexible function, for example a neural network.

## 3  Baselines: Variational Inference with VGAE

Before presenting our semi-implicit graph variational auto-encoder (SIG-VAE), we first introduce two baseline methods that directly combines SIVI and NF with VGAE.

**SIVI-VGAE.** To address **Problem 1** while well characterizing the posterior with modeling flexibility in the VGAE framework, the naive solution is to take the semi-implicit variational distribution in

SIVI for modeling latent variables in VGAE, following the hierarchical formulation

$$\mathbf{Z} \sim q(\mathbf{Z} \,|\, \psi), \qquad \psi \sim q_{\boldsymbol{\phi}}(\psi \,|\, \mathbf{X}, \mathbf{A}), \tag{3}$$

by introducing the implicit prior distribution parametrized by $\psi$, which can be sampled from the reparametrizable $q_{\boldsymbol{\phi}}(\psi \,|\, \mathbf{X}, \mathbf{A})$. Such a hierarchical semi-implicit construct not only leads to flexible mixture modeling of the posterior but also enables efficient model inference, for example, with $\phi$ being parameterized by deep neural networks. In this framework, the features from multiple layers of GNNs can be aggregated and then transformed via multiple fully connected layers after being concatenated by random noise to derive the posterior distribution for each node separately. More specifically, SIVI-VGAE injects random noise at $C$ different stochastic fully connected layers for each node independently:

$$\mathbf{h}_u = \mathrm{GNN}_u(\mathbf{A}, \mathrm{CONCAT}(\mathbf{X}, \mathbf{h}_{u-1})), \quad \text{for } u = 1, \dots, L, \ \mathbf{h}_0 = \mathbf{0}$$

$$\boldsymbol{\ell}_t^{(i)} = \boldsymbol{T}_t(\boldsymbol{\ell}_{t-1}^{(i)}, \boldsymbol{\epsilon}_t, \mathbf{h}_L^{(i)}), \quad \text{where } \boldsymbol{\epsilon}_t \sim q_t(\boldsymbol{\epsilon}) \text{ for } t = 1, \dots, C, \ \boldsymbol{\ell}_0^{(i)} = \mathbf{0}$$

$$\boldsymbol{\mu}_i(\mathbf{A}, \mathbf{X}) = \boldsymbol{g}_{\mu}(\boldsymbol{\ell}_C^{(i)}, \mathbf{h}_L^{(i)}), \ \ \boldsymbol{\Sigma}_i(\mathbf{A}, \mathbf{X}) = \boldsymbol{g}_{\Sigma}(\boldsymbol{\ell}_C^{(i)}, \mathbf{h}_L^{(i)}),$$

$$q(\mathbf{Z} \,|\, \mathbf{A}, \mathbf{X}, \boldsymbol{\mu}, \boldsymbol{\Sigma}) = \prod_{i=1}^{N} q(\mathbf{z}_i \,|\, \mathbf{A}, \mathbf{X}, \boldsymbol{\mu}_i, \boldsymbol{\Sigma}_i), \quad q(\mathbf{z}_i \,|\, \mathbf{A}, \mathbf{X}, \boldsymbol{\mu}_i, \boldsymbol{\Sigma}_i) = \mathcal{N}(\boldsymbol{\mu}_i(\mathbf{A}, \mathbf{X}), \boldsymbol{\Sigma}_i(\mathbf{A}, \mathbf{X})),$$

where $\boldsymbol{T}_t, \boldsymbol{g}_{\mu}$, and $\boldsymbol{g}_{\sigma}$ are all deterministic neural networks, $i$ is the node index, $L$ is the number of GNN layers, and $\boldsymbol{\epsilon}_t$ is random noise drawn from the distribution $q_t$. Note that in the equations above, GNN is any type of existing graph neural networks, such as graph convolutional neural network (GCN) [19], GCN with Chebyshev filters [13], GraphSAGE [15], jumping knowledge (JK) networks [36], and graph isomorphism network (GIN) [37]. Given the $\mathrm{GNN}_L$ output $\mathbf{h}_L$, $\boldsymbol{\mu}_i(\mathbf{A}, \mathbf{X})$ and $\boldsymbol{\Sigma}_i(\mathbf{A}, \mathbf{X})$ are now random variables rather than following vanilla VAE to assume deterministic values. In this way, however, the constructed implicit distributions may not capture the dependency between neighboring nodes completely. Note that we consider SIVI-VGAE as a naive version of our proposed SIG-VAE (and call it as **Naive SIG-VAE** in the rest of the paper), which is specifically designed with neighborhood sharing to capture complex dependency structures in networks, as detailed in the next section. Please also note that the first layer of SIVI can be integrated with NF rather than simple Gaussian. We leave that for future study.

**NF-VGAE.** It is also possible to enable VGAE model flexibility by other existing variational inference methods, for example using NF. However, NF requires deterministic transform functions whose Jacobians shall be easy to compute, which limits the flexibility when considering complex dependency structures in graph analytic tasks. We indeed have constructed a non-Gaussian VGAE, i.e. NF-based variational graph auto-encoder (NF-VGAE) as follows

$$\mathbf{h}_u = \mathrm{GNN}_u(\mathbf{A}, \mathrm{CONCAT}(\mathbf{X}, \mathbf{h}_{u-1})), \quad \text{for } u = 1, \dots, L, \ \mathbf{h}_0 = \mathbf{0} \tag{4}$$

$$\boldsymbol{\mu}(\mathbf{A}, \mathbf{X}) = \mathrm{GNN}_{\mu}(\mathbf{A}, \mathrm{CONCAT}(\mathbf{X}, \mathbf{h}_L)), \quad \boldsymbol{\Sigma}(\mathbf{A}, \mathbf{X}) = \mathrm{GNN}_{\Sigma}(\mathbf{A}, \mathrm{CONCAT}(\mathbf{X}, \mathbf{h}_L)),$$

$$q_0(\mathbf{Z}^{(0)} \,|\, \mathbf{A}, \mathbf{X}) = \prod_{i=1}^{N} q_0(\mathbf{z}_i^{(0)} \,|\, \mathbf{A}, \mathbf{X}), \quad \text{with} \quad q_0(\mathbf{z}_i^{(0)} \,|\, \mathbf{A}, \mathbf{X}) = \mathcal{N}(\boldsymbol{\mu}_i, \mathrm{diag}(\boldsymbol{\sigma}_i^2)),$$

$$q_K(\mathbf{Z}^{(K)} \,|\, \mathbf{A}, \mathbf{X}) = \prod_{i=1}^{N} q_0(\mathbf{z}_i^{(K)} | \mathbf{A}, \mathbf{X}), \quad \ln(q_K(\mathbf{z}_i^{(K)} \,|\, -)) = \ln(q_0(\mathbf{z}_i^{(0)})) - \sum_k \ln|\det \frac{\partial f_k}{\partial \mathbf{z}_i^{(k)}}|,$$

where the posterior distribution $q_K(\mathbf{Z}^{(K)} | \mathbf{A}, \mathbf{X})$ is obtained by successively transforming a Gaussian random variable $\mathbf{Z}^{(0)}$ with distribution $q_0$ through a chain of $K$ invertible differentiable transformations $f_k : \mathbb{R}^d \to \mathbb{R}^d$. We will further discuss this in the supplementary material. NF-VGAE is a two-step inference method that 1) starts with Gaussian random variables and then 2) transforms them through a series of invertible mappings. We emphasize again that in NF-VGAE, the **GNN** output layers are deterministic without neighborhood distribution sharing due to the deterministic nature of the initial density parameters in $q_0$.

## 4 Semi-implicit graph variational auto-encoder (SIG-VAE)

While the above two models are able to approximate more flexible and complex posterior, such trivial combinations may fail to fully exploit graph dependency structure because they are not capable of propagating uncertainty between neighboring nodes. To enable effective uncertainty propagation, which is the essential factor to capture complex posteriors with graph data, we develop a *carefully*

designed generative model, SIG-VAE, to better integrate variational inference and VGAE with a natural neighborhood sharing scheme.

To have tractable posterior inference, we construct SIG-VAE using a hierarchy of multiple stochastic layers. Specifically, the first stochastic layer $q(\mathbf{Z} \,|\, \mathbf{X}, \mathbf{A})$ is reparameterizable and has an analytic probability density function. The layers added after are reparameterizable and computationally efficient to sample from. More specifically, we adopt a hierarchical encoder in SIG-VAE that injects random noise at $L$ different stochastic layers:

$$\mathbf{h}_u = \mathrm{GNN}_u(\mathbf{A}, \mathrm{CONCAT}(\mathbf{X}, \boldsymbol{\epsilon}_u, \mathbf{h}_{u-1})), \quad \text{where } \boldsymbol{\epsilon}_u \sim q_u(\boldsymbol{\epsilon}) \text{ for } u = 1, \dots, L, \ \mathbf{h}_0 = \mathbf{0} \quad (5)$$

$$\boldsymbol{\mu}(\mathbf{A}, \mathbf{X}) = \mathrm{GNN}_\mu(\mathbf{A}, \mathrm{CONCAT}(\mathbf{X}, \mathbf{h}_L)), \quad \boldsymbol{\Sigma}(\mathbf{A}, \mathbf{X}) = \mathrm{GNN}_\Sigma(\mathbf{A}, \mathrm{CONCAT}(\mathbf{X}, \mathbf{h}_L)), \quad (6)$$

$$q(\mathbf{Z} \,|\, \mathbf{A}, \mathbf{X}, \boldsymbol{\mu}, \boldsymbol{\Sigma}) = \prod_{i=1}^{N} q(\mathbf{z}_i \,|\, \mathbf{A}, \mathbf{X}, \boldsymbol{\mu}_i, \boldsymbol{\Sigma}_i), \quad q(\mathbf{z}_i \,|\, \mathbf{A}, \mathbf{X}, \boldsymbol{\mu}_i, \boldsymbol{\Sigma}_i) = \mathcal{N}(\boldsymbol{\mu}_i(\mathbf{A}, \mathbf{X}), \boldsymbol{\Sigma}_i(\mathbf{A}, \mathbf{X})).$$

Note that in the equations above $\boldsymbol{\mu}$ and $\boldsymbol{\Sigma}$ are random variables and thus $q(\mathbf{Z} \,|\, \mathbf{X}, \mathbf{A})$ is not necessarily Gaussian after marginalization; $\boldsymbol{\epsilon}_u$ is $N$-dimensional random noise drawn from a distribution $q_u$; and $q_u$ is chosen such that the samples drawn from it are the same type as $\mathbf{X}$, for example if $\mathbf{X}$ is categorical, Bernoulli is a good choice for $q_u$. By concatenating the random noise and node attributes, the output of GNNs are random variables rather than deterministic vectors. Their expressive power is inherited in SIG-VAE to go beyond Gaussian, exponential family, or von Mises-Fisher [11] posterior distributions for the derived latent representations.

In SIG-VAE, when inferring each node's latent posterior, we incorporate the distributions of the neighboring nodes, better capturing graph dependency structure than sharing determin- istic features from GNNs. More specifically, the input to our model at stochastic layer $u$ is $\mathrm{CONCAT}(\mathbf{X}, \boldsymbol{\epsilon}_u)$ so that the outputs of the sub- sequent stochastic layers give mixing distribu- tions by integrating information from neighbor- ing nodes (Fig. 1). The flexibility of SIG-VAE directly working on the stochastic distribution parameters in (5-6) allows neighborhood sharing to achieve better performance in graph analytic

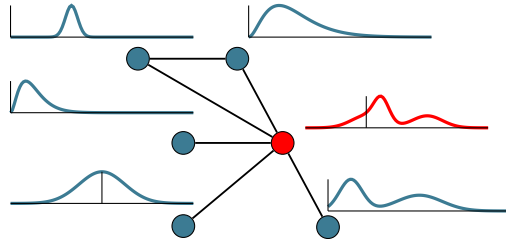

Figure 1: SIG-VAE diffuses the distributions of the neighboring nodes, which is more informative than shar- ing deterministic features, to infer each node's latent distribution.

tasks. We argue that the uncertainty propagation in our *carefully* designed SIG-VAE, which is the an outcome of using GNNs and adding noise in the input in equations (5-6), is the key factor in capturing more faithful and complex posteriors. Note that (5) is different from the NF-VAE construction (3), where the GNN output layers are deterministic. Through experiments, we show that this uncertainty neighborhood sharing is key for SIG-VAE to achieve superior graph analysis performance.

We further argue that increasing the flexibility of variational inference is not enough to better model real-world graph data as the optimal solution of the generative model does not change. In SIG-VAE, the Bernoulli-Poisson link [41] is adopted for the decoder to further increase the expressiveness of the generative model. Potential extensions with other decoders can be integrated with SIG-VAE if needed. Let $A_{i,j} = \delta(m_{ij} > 0)$, $m_{ij} \sim \mathrm{Poisson}\big( \exp(\sum_{k=1}^{l} r_k z_{ik}\, z_{jk}) \big)$, and hence

$$p(\mathbf{A} \,|\, \mathbf{Z}, \mathbf{R}) = \prod_{i=1}^{N} \prod_{j=1}^{N} p(A_{i,j} \,|\, \mathbf{z}_i, \mathbf{z}_j, \mathbf{R}), \quad p(A_{i,j} = 1 \,|\, \mathbf{z}_i, \mathbf{z}_j, \mathbf{R}) = 1 - e^{-\exp\left(\sum_{k=1}^{\mathscr{L}} r_k z_{ik}\, z_{jk}\right)}, \ (7)$$

where $\mathbf{R} \in \mathbb{R}_+^{\mathscr{L} \times \mathscr{L}}$ is a diagonal matrix with diagonal elements $r_k$.

## 4.1 Inference

To derive the ELBO for model inference in SIG-VAE, we must take into account the fact that $\psi$ has to be drawn from a distribution. Hence, the ELBO moves beyond the simple VGAE as

$$\mathcal{L} = -\mathbf{KL}(\mathbb{E}_{\psi \sim q_\phi(\psi \,|\, \mathbf{X}, \mathbf{A})}[q(\mathbf{Z} \,|\, \psi)] \,\|\, p(\mathbf{Z})) + \mathbb{E}_{\psi \sim q_\phi(\psi \,|\, \mathbf{X}, \mathbf{A})}[\mathbb{E}_{\mathbf{Z} \sim q(\mathbf{Z} \,|\, \psi)}[\log p(\mathbf{A} \,|\, \mathbf{Z})]], \quad (8)$$

where $h_\phi$ is defined in (1). The marginal probability density function $h_\phi(\mathbf{Z}|\mathbf{X}, \mathbf{A})$ is often intractable, so the Monte Carlo estimation of the ELBO, $\mathcal{L}$, is prohibited. To address this issue and infer

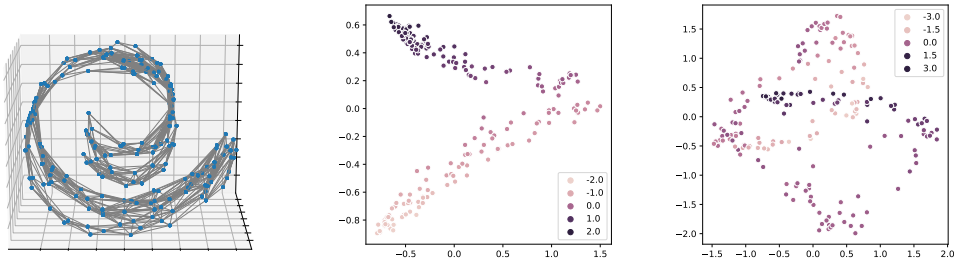

Figure 2: Swiss roll graph (left) and its latent representation using SIG-VAE (middle) and VGAE (right). The latent representations (middle and right) are heat maps in $\mathbb{R}^3$. We expect that the embedding of the Swiss roll graph with inner-product decoder to be a curved plane in $\mathbb{R}^3$, which is clearly captured better by SIG-VAE.



Figure 3: Latent representation distributions of five example nodes from the Swiss roll graph using SIG-VAE (blue) and VGAE (red). SIG-VAE clearly infers more complex distributions that can be multi-modal, skewed, and with sharp and steep changes. This helps SIG-VAE to better represent the nodes in the latent space.

variational parameters of SIG-VAE, we can derive a lower bound for the ELBO as follows (see the supplementary material for more details)

$$\underline{\mathcal{L}} = -\mathbb{E}_{\psi \sim q_\phi(\psi \,|\, \mathbf{X}, \mathbf{A})}[\mathbf{KL}(q(\mathbf{Z} \,|\, \psi) \,\|\, p(\mathbf{Z}))] + \mathbb{E}_{\psi \sim q_\phi(\psi \,|\, \mathbf{X}, \mathbf{A})}\big[\mathbb{E}_{\mathbf{Z} \sim q(\mathbf{Z} \,|\, \psi)}[\log p(\mathbf{A} \,|\, \mathbf{Z})]\big] \leq \mathcal{L}.$$

Further implementation details and the derivation of the surrogate ELBO can be found in the supplementary material.

# 5 Experiments

We test the performances of SIG-VAE on different graph analytic tasks: 1) interpretability of SIG-VAE compared to VGAE, 2) link prediction in various real-world graph datasets including graphs with node attributes and without node attributes, 3) graph generation, 4) node classification in the citation graphs with labels. In all of the experiments, GCN [19] is adopted for all the GNN modules in SIG-VAE, Naive SIG-VAE, and NF-VGAE, implemented in Tensorflow [1]. The PyGSP package [12] is used to generate synthetic graphs. Implementation details for all the experiments, together with graph data statistics, can be found in the supplementary material.

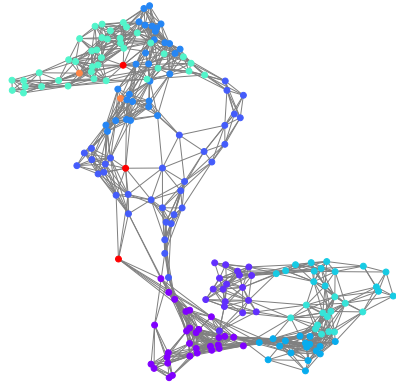

Figure 4: The nodes with multi-modal posteriors (red nodes) reside between different communities in Swiss Roll graph.

## 5.1 Interpretable latent representations

We first demonstrate the expressiveness of SIG-VAE by illustrating the approximated variational distributions of node latent representations. We show that SIG-VAE captures the graph structure better and has a more interpretable embedding than VGAE on a generated **Swiss roll** graph with 200 nodes and 1244 edges (Fig. 2). In order to provide a fair comparison, both models share an identical implementation with the inner-product decoder and same number of parameters. We simply consider the identity matrix $\mathcal{I}_N$ as node attributes and choose the latent space dimension to be three in this experiment. This graph has a simple plane like structure. As the inner-product decoder assumes that the information is embedded in the *angle* between latent vectors, we expect that the node embedding to map nodes of the Swiss roll graph into a curve in the latent space. As we can

Table 1: Link prediction performance in networks with node attributes.

| Method | Cora | | Citeseer | | Pubmed | |
|---|---|---|---|---|---|---|
| | AUC | AP | AUC | AP | AUC | AP |
| SC [31] | $84.6 \pm 0.01$ | $88.5 \pm 0.00$ | $80.5 \pm 0.01$ | $85.0 \pm 0.01$ | $84.2 \pm 0.02$ | $87.8 \pm 0.01$ |
| DW [25] | $83.1 \pm 0.01$ | $85.0 \pm 0.00$ | $80.5 \pm 0.02$ | $83.6 \pm 0.01$ | $84.4 \pm 0.00$ | $84.1 \pm 0.00$ |
| GAE [18] | $91.0 \pm 0.02$ | $92.0 \pm 0.03$ | $89.5 \pm 0.04$ | $89.9 \pm 0.05$ | $96.4 \pm 0.00$ | $96.5 \pm 0.00$ |
| VGAE [18] | $91.4 \pm 0.01$ | $92.6 \pm 0.01$ | $90.8 \pm 0.02$ | $92.0 \pm 0.02$ | $94.4 \pm 0.02$ | $94.7 \pm 0.02$ |
| $\mathcal{S}$-VGAE [11] | $94.10 \pm 0.1$ | $94.10 \pm 0.3$ | $94.70 \pm 0.2$ | $95.20 \pm 0.2$ | $96.00 \pm 0.1$ | $96.00 \pm 0.1$ |
| SEAL [40] | $90.09 \pm 0.1$ | $83.01 \pm 0.3$ | $83.56 \pm 0.2$ | $77.58 \pm 0.2$ | $96.71 \pm 0.1$ | $90.10 \pm 0.1$ |
| G2G [9] | $92.10 \pm 0.9$ | $92.58 \pm 0.8$ | $95.32 \pm 0.7$ | $95.57 \pm 0.7$ | $94.28 \pm 0.3$ | $93.38 \pm 0.5$ |
| **NF-VGAE** | $92.42 \pm 0.6$ | $93.08 \pm 0.5$ | $91.76 \pm 0.3$ | $93.04 \pm 0.8$ | $96.59 \pm 0.3$ | $96.68 \pm 0.4$ |
| **Naive SIG-VAE** | $93.97 \pm 0.5$ | $93.29 \pm 0.4$ | $94.25 \pm 0.8$ | $93.60 \pm 0.9$ | $96.53 \pm 0.7$ | $96.01 \pm 0.5$ |
| **SIG-VAE** (IP) | $\mathbf{94.37} \pm 0.1$ | $\mathbf{94.41} \pm 0.1$ | $\mathbf{95.90} \pm 0.1$ | $\mathbf{95.46} \pm 0.1$ | $\mathbf{96.73} \pm 0.1$ | $\mathbf{96.67} \pm 0.1$ |
| **SIG-VAE** | $\mathbf{96.04} \pm 0.04$ | $\mathbf{95.82} \pm 0.06$ | $\mathbf{96.43} \pm 0.02$ | $\mathbf{96.32} \pm 0.02$ | $\mathbf{97.01} \pm 0.07$ | $\mathbf{97.15} \pm 0.04$ |

see in Fig. 2, SIG-VAE derives a clearly more interpretable planar latent structure than VGAE. We also show the posterior distributions of five randomly selected nodes from the graph in Fig. 3. As we can see, SIG-VAE is capable of inferring complex distributions. The inferred distributions can be multi-modal, skewed, non-symmetric, and with sharp and steep changes. These complex distributions help the model to get a more realistic embedding capturing the intrinsic graph structure. To explain why multi-modality may arise, we used Asynchronous Fluid [24] to visualize the Swiss Roll graph by highlighting detected communities with different colors in Fig. 4. Note that we used a different layout from the one in Fig. 2(a) to better visualize the communities in the graph. The three red (two orange) nodes are the nodes with multi-modal (skewed) distributions in Fig. 3. These nodes with multi-modal posteriors reside between different communities; hence, with a probability, they could be assigned to multiple communities. The supplementary material contains additional results and discussions with a **torus** graph, with similar observations.

## 5.2 Accurate link prediction

We further conduct extensive experiments for link prediction with various real-world graph datasets. Our results show that SIG-VAE significantly outperforms well-known baselines and state-of-the-art methods in all benchmark datasets. We consider two types of datasets, i.e., datasets with node attributes and datasets without attributes. We preprocess and split the datasets as done in Kipf and Welling [18] with validation and test sets containing 5% and 10% of network links, respectively. We learn the model parameters for 3500 epochs with the learning rate 0.0005 and the validation set used for early stopping. The latent space dimension is set to 16. The hyperparameters of SIG-VAE, Naive SIG-VAE, and NF-VGAE are the same for all the datasets. For fair comparison, all methods have the similar number of parameters as the default VGAE. The supplementary material contains further implementation details. We measure the performance by average precision (AP) and area under the ROC curve (AUC) based on 10 runs on a test set of previously removed links in these graphs.

**With node attributes.** We consider three graph datasets with node attribbutes—Citeseer, Cora, and Pubmed [28]. The number of node attributes for these dataset are 3703, 1433, and 500 respectively. Other statistics of the datasets are summarized in the supplement Table 1. We compare the results of SIG-VAE, Naive SIG-VAE, and NF-VGAE with six state-of-the-art methods, including spectral clustering (SC), DeepWalk (DW) [25] , GAE [18], VGAE [18], $\mathcal{S}$-VGAE [11], and SEAL [40]. The inner-product decoder is also adopted in SIG-VAE to clearly demonstrate the advantages of the semi-implicit hierarchical variational distribution for the encoder.

We use the same hyperparameters for the competing methods as stated in [40, 18, 11]. As we can see in Table 1, SIG-VAE shows significant improvement in terms of both AUC and AP over state-of-the-art methods. Note the standard deviation of SIG-VAE is also smaller compared to other methods, indicating stable semi-implicit variational inference. Compared to the baseline VGAE, more flexible posterior in three proposed methods SIGVAE (with both inner-product and Bernoulli-Poisson link decoders), Naive SIG-VAE, and NF-VGAE can clearly improve the link prediction accuracy. This suggests that the Gaussian assumption does not hold for these graph structured data. The performance improvement of SIG-VAE with inner-product decoder (IP) over Naive SIG-VAE and NF-VGAE clearly demonstrates the advantages of neighboring node sharing, especially in the smaller graphs. Even for the large graph Pubmed, on which VGAE performs similar to $\mathcal{S}$-VGAE, our SIG-VAE still achieves the highest link prediction accuracy, showing the importance of all modeling components

Table 2: AUC and AP of link prediction in networks without node attributes. * indicates that the numbers are reported from Zhang and Chen [40]. The supplementary material contains the complete result tables with standard deviation values.

| Metrics | Data | MF* | SBM* | N2V* | LINE* | SC* | GAE | VGAE* | SEAL* | G2G | NF-VGAE | N-SIG-VAE | SIG-VAE(IP) | SIG-VAE |
|---|---|---|---|---|---|---|---|---|---|---|---|---|---|---|
| **AUC** | **USAir** | 94.08 | 94.85 | 91.44 | 81.47 | 74.22 | 93.09 | 89.28 | 97.09 | 92.17 | 95.74 | 94.22 | **97.56** | 94.52 |
| | **NS** | 74.55 | 92.30 | 91.52 | 80.63 | 89.94 | 93.14 | 94.04 | 97.71 | 98.18 | 98.38 | 98.00 | **98.75** | **99.17** |
| | **Yeast** | 90.28 | 91.41 | 93.67 | 87.45 | 93.25 | 93.74 | 93.88 | 97.20 | 97.34 | 97.86 | 93.36 | **98.11** | **98.32** |
| | **Power** | 50.63 | 66.57 | 76.22 | 55.637 | 91.78 | 72.21 | 71.20 | 84.18 | 91.35 | 94.61 | 93.67 | **95.04** | **96.23** |
| | **Router** | 78.03 | 85.65 | 65.46 | 67.15 | 68.79 | 55.73 | 61.51 | 95.68 | 85.98 | 93.56 | 92.66 | **95.94** | **96.13** |
| **AP** | **USAir** | 94.36 | 95.08 | 89.71 | 79.70 | 78.07 | 95.14 | 89.27 | 95.70 | 90.22 | 96.27 | 94.48 | **97.50** | 94.95 |
| | **NS** | 78.41 | 92.13 | 94.28 | 85.17 | 90.83 | 95.26 | 95.83 | 98.12 | 97.43 | 98.52 | 97.83 | **98.53** | **99.24** |
| | **Yeast** | 92.01 | 92.73 | 94.90 | 90.55 | 94.63 | 95.34 | 95.19 | 97.95 | 97.83 | 98.18 | 94.24 | **97.97** | **98.41** |
| | **Power** | 53.50 | 65.48 | 81.49 | 56.66 | 91.00 | 77.13 | 75.91 | 86.69 | 92.29 | 95.76 | 93.80 | **96.50** | **97.28** |
| | **Router** | 82.59 | 84.67 | 68.66 | 71.92 | 73.53 | 67.50 | 70.36 | 95.66 | 86.28 | 95.88 | 92.80 | **94.94** | **96.86** |

in the proposed method including non-Gaussian posterior, using neighborhood distribution, and the sparse Bernoulli-Poisson link decoder.

**Without node attributes.** We further consider five graph datasets without node attributes—USAir, NS [22], Router [29], Power [34] and Yeast [32]. The data statistics are summarized in the supplement Table 1. We compare the performance of our models with seven competing state-of-the-art methods including matrix factorization (MF), stochastic block model (SBM) [3], node2vec (N2V) [14], LINE [30], spectral clustering (SC), VGAE [18], $\mathcal{S}$-VGAE [11], and SEAL [40].

For baseline methods, we use the same hyperparameters as stated in Zhang et al. [40]. For datasets without node attributes, we use a two-stage learning process for SIG-VAE. First, the embedding of each node is learned in the 128-dimensional latent space while injecting 5-dimensional Bernoulli noise to the system. Then the learned embedding is taken as node features for the second stage to learn 16 dimensional embedding while injecting 64-dimensional noise to SIG-VAE. Through empirical experiments, we found that this two-stage learning converges faster than end-to-end learning. We follow the same procedure for Naive SIG-VAE and NF-VGAE.

As we can see in Table 2, SIG-VAE again shows the consistent superior performance compared to the competing methods, especially over the baseline VGAE, in both AUC and AP. It is interesting to note that, while the proposed Berhoulli-Poisson decoder works well for sparser graphs, especially NS and Router datasets, SIG-VAE with inner-product decoder shows superior performance for the USAir graph which is much denser. Compared to the baseline VGAE, both Naive SIG-VAE and NF-VGAE improve the results with a large margin in both AUC and AP, showing the benefits of more flexible posterior. Comparing SIG-VAE with two other flexible inference methods shows not only SIG-VAE is not restricted to the Gaussian assumption, which is not a good fit for link prediction with the inner-product decoder [11], but also it is able to model flexible posterior considering graph topology. The results for the link prediction of the Power graph clearly magnifies this fact as SIG-VAE improves the accuracy by 34% compared to VGAE. The supplementary material contains the results with standard deviation values over different runs, showing the stability again.

Ablation studies have also been run to evaluate SIG-VAE with inner-product decoder in link prediction for citation graphs without using node attributes. The [AUC, AP] are [91.14, 90.99] for Cora and [88.72, 88.24] for Citeseer, lower than the values from SIG-VAE with attributes in Table 1 but are still competitive against existing methods (even with node attributes), showing the ability of SIG-VAE of utilizing graph structure. While some of the methods, like SEAL, work well for graphs without node attributes and some of others, like VGAE, get good performance for graphs with node attributes, SIG-VAE consistently achieves superior performance in both types of datasets. This is due to the fact that SIG-VAE can learn implicit distributions for nodes, which are very powerful in capturing graph structure even without any node attributes.

## 5.3   Graph generation

To further demonstrate the flexibility of SIG-VAE as a generative model, we have used the inferred embedding representations to generate new graphs. For example, SIG-VAE infers network parameters for Cora whose density and average clustering coefficients are 0.00143 and 0.24, respectively. Using the inferred posterior and learned decoder, a new graph is generated with corresponding $r_k$ to see if its graph statistics are close to the original ones. Please note that we have shrunk inferred $r_k$'s smaller than 0.01 to 0. The density and average clustering coefficients of this generated graph based on SIG-VAE are 0.00147 and 0.25, respectively, which are very close to the original graph. We also generate new graphs based on SIG-VAE with the inner-product decoder and VGAE. The density and

average clustering coefficients of the generated graphs based on SIG-VAE (IP) and VGAE are same, i.e. 0.1178 and 0.49, respectively, showing the inner-product decoder may not be a good choice for sparse graphs. The supplementary material includes more examples.

### 5.4 Node classification & graph clustering

We also have applied SIG-VAE for node classification on citation graphs with labels by modifying the loss function to include graph reconstruction and semi-supervised classification terms. Results are summarized in Table 3. Our model exhibits strong generalization properties, highlighted by its competitive performance compared to the state-of-the-art methods, despite not being trained specifically for this task. To show the robustness of SIG-VAE to missing edges, we randomly removed 10, 20, 50 and 70 (%) edges while keeping node attributes. The mean accuracy of 10 run for Cora (2 layers [32,16]) are 79.5, 78.7, 75.3 and 60.6, respectively. The supplementary material contains additional results and discussion for graph clustering, again without specific model tuning.

Table 3: Summary of results in terms of classification accuracy (in percent).

| Method | Cora | Citeseer | Pubmed |
|---|---|---|---|
| ManiReg [6] | 59.5 | 60.1 | 70.7 |
| SemiEmb [35] | 59.0 | 59.6 | 71.1 |
| LP [42] | 68.0 | 45.3 | 63.0 |
| DeepWalk [25] | 67.2 | 43.2 | 65.3 |
| ICA [20] | 75.1 | 69.1 | 73.9 |
| Planetoid [38] | 75.7 | 64.7 | 77.2 |
| GCN [19] | **81.5** | 70.3 | 79.0 |
| **SIG-VAE** | 79.7 | **70.4** | **79.3** |

SIG-VAE has demonstrated state-of-the-art performances in link prediction and comparable results on other tasks, clearly showing the potential of SIG-VAE on different graph analytic tasks.

## 6 Conclusion

Combining the advantages of semi-implicit hierarchical variational distribution and VGAE with a Bernoulli-Poisson link decoder, SIG-VAE is developed to enrich the representation power of the posterior distribution of node embedding given graphs so that both the graph structural and node attribute information can be best captured in the latent space. By providing a surrogate evidence lower bound that is asymptotically exact, the optimization problem for SIG-VAE model inference is amenable via stochastic gradient descent, without compromising the flexbility of its variational distribution. Our experiments with different graph datasets have shown the promising capability of SIG-VAE in a range of graph analysis applications with interpretable latent representations, thanks to the hierarchical construction that diffuses the distributions of neighborhood nodes in given graphs.

## 7 Acknowledgments

The presented materials are based upon the work supported by the National Science Foundation under Grants ENG-1839816, IIS-1848596, CCF-1553281, IIS-1812641 and IIS-1812699. We also thank Texas A&M High Performance Research Computing and Texas Advanced Computing Center for providing computational resources to perform experiments in this work.

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
