[Supplementary Material]

# Semi-Implicit Graph Variational Auto-Encoders: Supplementary Material

**Arman Hasanzadeh**[†][*], **Ehsan Hajiramezanali**[†][*], **Nick Duffield**[†], **Krishna Narayanan**[†], **Mingyuan Zhou**[‡], **Xiaoning Qian**[†]

† Department of Electrical and Computer Engineering, Texas A&M University
{armanihm, ehsanr, duffieldng, krn, xqian}@tamu.edu
‡ McCombs School of Business, The University of Texas at Austin
mingyuan.zhou@mccombs.utexas.edu

In this supplement, we first provide the detailed review of the related literature as well as the connection to our proposed work. Derivation of ELBO, dataset statistics, network setups, and implementation details of performance evaluation experiments for different graph analytic tasks are then presented with richer experimental results in addition to the ones discussed in the main text.

## 1 Related works

Variational graph auto-encoders (VGAE), proposed by Kipf and Welling [5], embed each node to a random variable in the latent space. VGAE, by extending the use of VAEs to graph structured data, is shown to be capable of learning interpretable latent representations for undirected graphs and getting competitive results in the link prediction task. However, the Gaussian assumption imposed on the variational distribution restricts the model flexibility when the true posterior distribution given a graph clearly violates the assumption. It also suffers from underestimating the variance of the posterior, which is a well-known issue of vanilla VAEs.

To better model graph data using variational distributions in VGAEs, Davidson et al. [2] proposes the hyperspherical VGAE ($\mathcal{S}$-VGAE), in which, instead of the Gaussian assumption for the posterior, the von Mises-Fisher distribution has been deployed. This assumption is not well-suited for all classes of graphs. For example, it has been proven that graphs with hierarchical tree-like structure have hyperbolic latent structures [7] which clearly cannot be represented well in a hyperspherical space. While $\mathcal{S}$-VGAE outperforms vanilla VGAE in some graphs including Cora and Citeseer in terms of link prediction accuracy, its performance will be degraded for more complex graphs such as Pubmed. On the other hand, changing the prior is not going to change the flexibility and optimal solution of the generative model, but will affect the tightness of the ELBO and hence how well the generative model parameters can be inferred. This shows the necessity to develop a variational graph auto-encoders that not only is capable of inferring more flexible posteriors to represent a broader range of graphs, but also is able to have more flexible decoder especially for the real-world sparse graphs.

In this paper, we propose to develop a hierarchical variational model to increase the expressiveness of the posterior distribution for each node in the latent space. While SIVI-VGAE and NF-VGAE can be used as a variational node embedding to effectively expand the variational posterior distribution family, SIG-VAE allows flexible implicit posteriors as well as exploitation of the neighbor dependency while maintaining simple optimization. We have further adopted a Bernoulli-Poisson link decoder to improve the flexibility of the generative model which has not been addressed in the previous studies.

---

[*]Both authors contributed equally.

## 2 Node embedding

Node embedding is to represent each node in a graph by a low-dimensional vector in a latent space. The geometric relations of vectors in the latent space reflect the probability of two corresponding nodes interacting with each other in the graph [4]. A good node embedding preserves node connectivity in graph as well as local neighborhood structures. More formally, node embedding can be formulated as follows.

**Node embedding.** Given a graph $G = (\mathcal{V}, \mathcal{E})$ where $\mathcal{V}$ is the set of nodes and $\mathcal{E}$ the set of edges, with the adjacency matrix $\mathbf{A}$, $\mathbf{X} \in \mathbb{R}^{N \times M}$ denoting $M$-dimensional node attributes for $N = |\mathcal{V}|$ nodes, and a function $s_G : \mathcal{V} \times \mathcal{V} \to \mathbb{R}$ measuring node similarity, find an encoder function, $\mathbf{ENC} : \mathbb{R}_+^{N \times N} \times \mathbb{R}^{N \times M} \to \mathbb{R}^l$, a decoder function, $\mathbf{DEC} : \mathbb{R}^l \times \mathbb{R}^l \to \mathbb{R}_+$, and a latent representation of nodes $\mathbf{Z} \in \mathbb{R}^{N \times l}$ such that

$$\mathbf{Z} = \mathbf{ENC}(\mathbf{A}, \mathbf{X}),$$

$$\hat{s}_{i,j} \triangleq \mathbf{DEC}(\mathbf{z}_i, \mathbf{z}_j),$$

where $\mathbf{z}_i$ corresponds to the embedding representation of node $v_i \in \mathcal{V}$. Optimal parameters of $\mathbf{ENC}$ and $\mathbf{DEC}$ functions can be derived by finding the solutions to the following optimization problem

$$\min_{\mathbf{ENC},\mathbf{DEC}} \quad \sum_{i=1}^{N} \mathbf{loss}(\hat{s}_{i,j}, s_G(v_i, v_j)),$$

where **loss** is a user-specified loss function based on the ultimate objective of network analysis.

Different node embedding methods vary in the choice of the **loss** function, $s_G$, **ENC**, **DEC** and the optimization algorithm. For example, in graph factorization (GF) method [1], $s_G$ is defined based on the adjacency matrix, i.e., $s_G(v_i, v_j) = A_{i,j}$; **loss** is the mean squared error; and the inner-product decoder is adopted, i.e., $\mathbf{DEC}(\mathbf{z}_i, \mathbf{z}_j) = \mathbf{z}_i^T \mathbf{z}_j$.

## 3 Variational inference with normalizing flows

To increase the expressive power of a probabilistic model, a simple but powerful idea is to transform the corresponding random variables with complex deterministic and/or stochastic mappings. To construct flexible, arbitrarily complex and scalable approximate posterior distributions, normalizing flow (NF) transforms a simple random variable through a sequence of invertible differentiable functions with tractable Jacobians. More specifically, NF uses an invertible, smooth mapping $f : \mathbb{R}^d \to \mathbb{R}^d$ to transform a random variable $z$ with distribution $q(z)$ to the resulting random variable $z' = f(z)$ with the distribution:

$$q(z') = q(z) \left| \det \frac{\partial f^{-1}}{\partial z'} \right| = q(z) \left| \det \frac{\partial f}{\partial z} \right|^{-1}. \tag{1}$$

One may apply a chain of $K$ transformations $f_k$ to obtain the density $q_K(z)$ from a random variable $z_0$ with distribution $q_0$ as:

$$\ln q_K(z_K) = \ln q_0(z_0) - \sum_k \ln \left| \det \frac{\partial f_k}{\partial z_k} \right|. \tag{2}$$

While normalizing flow helps to improve the model flexibility of the corresponding variational posterior, it requires the mapping to be deterministic and invertible, and the mixing distribution in the hierarchy to have an explicit density function. Removing these restrictions, there have been several recent attempts to define highly flexible variational posterior with implicit models. While an implicit variational distribution can be made highly flexible, it becomes necessary in each iteration to address the problem of density ratio estimation, which is often transformed into a problem related to learning generative adversarial networks [3]. SIVI addresses this issue by using an analytic conditional variational distribution which is not required to be reparameterizable.

## 4   SIG-VAE inference details

To derive the ELBO for model inference in SIG-VAE, we must take into account the fact that $\psi$ has to be drawn from a distribution. Hence, the ELBO moves beyond the simple VGAE as

$$
\begin{aligned}
\mathcal{L} &= -\mathbf{KL}(\mathbb{E}_{\psi \sim q_\phi(\psi \,|\, \mathbf{X}, \mathbf{A})}[q(\mathbf{Z} \,|\, \psi)] \,\|\, p(\mathbf{Z})) + \mathbb{E}_{\psi \sim q_\phi(\psi \,|\, \mathbf{X}, \mathbf{A})}[\mathbb{E}_{\mathbf{Z} \sim q(\mathbf{Z} \,|\, \psi)}[\log p(\mathbf{A} \,|\, \mathbf{Z})]] \\
&= \mathbb{E}_{\mathbf{Z} \sim h_\phi(\mathbf{Z} \,|\, \mathbf{X}, \mathbf{A})}\Big[\log \frac{p(\mathbf{A} \,|\, \mathbf{Z})p(\mathbf{Z})}{h_\phi(\mathbf{Z} \,|\, \mathbf{X}, \mathbf{A})}\Big].
\end{aligned}
\tag{3}
$$

Direct optimization of the ELBO in SIVI is not tractable [9], so the Monte Carlo estimation of the ELBO, $\mathcal{L}$, is prohibited. To address this issue, SIVI derives a lower bound for the ELBO and optimizes this lower bound instead of optimizing the ELBO itself, which is tractable and asymptotically equals to the ELBO. SIG-VAE requires $q(\mathbf{Z} \,|\, \psi)$ to be explicit, and also requires it to either be reparameterizable or the ELBO under $q(\mathbf{Z} \,|\, \psi)$ to be analytic, while $q_\phi(\psi \,|\, \mathbf{X}, \mathbf{A})$ is required to be reparameterizable but not necessarily explicit. This captures the idea that combining an explicit $q(\mathbf{Z} \,|\, \psi)$ with an implicit $q_\phi(\psi \,|\, \mathbf{X}, \mathbf{A})$ is as powerful as needed, but makes the computation tractable.

Following Yin and Zhou [9], we can derive a lower bound for the ELBO as follows

$$
\begin{aligned}
\underline{\mathcal{L}} &= \mathbb{E}_{\psi \sim q_\phi(\psi \,|\, \mathbf{X}, \mathbf{A})}\Big[\mathbb{E}_{\mathbf{Z} \sim q(\mathbf{Z} \,|\, \psi)}\Big[\log \Big(\frac{p(\mathbf{A}|\mathbf{Z})p(\mathbf{Z})}{q(\mathbf{Z} \,|\, \psi)}\Big)\Big]\Big] \\
&= -\mathbb{E}_{\psi \sim q_\phi(\psi \,|\, \mathbf{X}, \mathbf{A})}[\mathbf{KL}(q(\mathbf{Z} \,|\, \psi) \,\|\, p(\mathbf{Z}))] + \mathbb{E}_{\psi \sim q_\phi(\psi \,|\, \mathbf{X}, \mathbf{A})}\big[\mathbb{E}_{\mathbf{Z} \sim q(\mathbf{Z} \,|\, \psi)}[\log p(\mathbf{A} \,|\, \mathbf{Z})]\big] \leq \mathcal{L}.
\end{aligned}
$$

This can be proved based on the first theorem in Yin and Zhou [9], which shows

$$
\mathbf{KL}(\mathbb{E}_{\psi \sim q_\phi(\psi \,|\, \mathbf{X}, \mathbf{A})}[q(\mathbf{Z} \,|\, \psi)] \,\|\, p(\mathbf{Z})) \leq \mathbb{E}_{\psi \sim q_\phi(\psi \,|\, \mathbf{X}, \mathbf{A})}[\mathbf{KL}(q(\mathbf{Z} \,|\, \psi) \,\|\, p(\mathbf{Z}))].
$$

Unlike $\mathcal{L}$, a Monte Carlo estimation of $\underline{\mathcal{L}}$ only requires $q_\phi(\mathbf{Z} \,|\, \psi)$ to have an analytic density functions and $q_\phi(\psi \,|\, \mathbf{X}, \mathbf{A})$ to be convenient to sample from.

Directly optimizing $\underline{\mathcal{L}}$ without early stopping could lead to a point mass density as $q_\phi(\psi \,|\, \mathbf{X}, \mathbf{A})$. This degenerates SIG-VAE to the vanilla VGAE. To avoid degeneracy, a regularization term can be added to $\underline{\mathcal{L}}$. Assume that $K$ samples are drawn from $q_\phi(\psi \,|\, \mathbf{X}, \mathbf{A})$ denoted by $\{\psi^{(i)}\}_{i=1}^K$. We define a regularized lower bound as $\underline{\mathcal{L}}_K = \underline{\mathcal{L}} + B_K$ where

$$
B_K = \mathbb{E}_{\psi, \psi^{(1)}, \dots, \psi^{(K)} \sim q_\phi(\psi \,|\, \mathbf{X}, \mathbf{A})}[\mathbf{KL}(q(\mathbf{A} \,|\, \psi) \,\|\, \tilde{h}_K(\mathbf{Z}))],
$$

and

$$
\tilde{h}_K(\mathbf{Z})) = \frac{q_\phi(\psi \,|\, \mathbf{X}, \mathbf{A}) + \sum_{k=1}^K q_\phi(\psi^{(k)} \,|\, \mathbf{X}, \mathbf{A})}{K + 1}.
$$

It has been proved by Molchanov et al. [6] that $\underline{\mathcal{L}}_K$ is a monotonic lower bound of the ELBO, satisfying $\underline{\mathcal{L}}_K \leq \underline{\mathcal{L}}_{K+1} \leq \mathcal{L}$. Therefore, setting $K$ to zero means that $\underline{\mathcal{L}}_0 = \underline{\mathcal{L}}$, and as $K$ goes to infinity $\underline{\mathcal{L}}$ converges to the exact ELBO, i.e., $\lim_{K \to \infty} \underline{\mathcal{L}}_K = \mathcal{L}$.

## 5   Graph dataset details

Table 1 provides the detailed statistics of the graph datasets used in our experiments.

Table 1: Graph dataset statistics.

| Dataset | Type | Nodes | Edges |
|---|---|---|---|
| **Cora** | Citation | 2,708 | 5,429 |
| **Citeseer** | Citation | 3,327 | 4,732 |
| **Pubmed** | Citation | 19,717 | 44,338 |
| **USAir** | Transportation | 332 | 2,126 |
| **NS** | Collaboration | 1,589 | 2,742 |
| **Router** | Internet | 5,022 | 6,258 |
| **Power** | Energy | 4,941 | 6,594 |
| **Yeast** | Protein | 2,375 | 11,693 |

# 6 Experimental setups and hyperparameter tuning

**Interpretable latent representations experiments.** In these experiments, the code provided by Kipf and Welling [5] is used to derive the embedding for VGAE. The size of the first hidden layer of VGAE is 256 and the size of the output layer is 3. For SIG-VAE, two stochastic layers with sizes equal to [32, 32] and an additional GCN layer of size 16 are used to model the $\boldsymbol{\mu}$. The dimension of injected standard Gaussian noises $[\epsilon_1, \epsilon_2]$ are [32, 32]. Covariance matrix $\boldsymbol{\Sigma}$ is deterministic and is inferred through two layers of GCNs with sizes equal to [32, 16]. To remove the effect of decoder, we consider the inner-product decoder for this set of experiments.

**Link prediction with node attributes** For SIG-VAE, we use a stochastic layer with size equal to 32 and an additional GCN layer of size 16 is used to model $\boldsymbol{\mu}$. The dimension of injected Bernoulli noise $\epsilon$ for the stochastic layer is 64. For SIVI-VGAE, we use two GCN layers with sizes equal to [32, 16] followed by a fully connected layers with size 16 to infer $\boldsymbol{\mu}$. We inject 64-dimensional Bernoulli noise to the fully connected layer. We implement NF-VGAE by extending VGAE (two GCN layers with sizes equal to [32, 16]) with invertible linear-time transformations of length 4 to keep its number of parameters close to the competing methods. We learn the model parameters for 3500 epochs with the learning rate 0.0005 and the validation set used for early stopping.

**Link prediction without node attributes.** For SIG-VAE, we use a stochastic layer with size equal to 32 and an additional GCN layer of size 16 is used to model $\boldsymbol{\mu}$. The dimensions of injected Bernoulli noise $\epsilon$ is 32. For SIVI-VGAE, we use two GCN layers with sizes equal to [32, 16] followed by a fully connected layer with sizes 16 to infer $\boldsymbol{\mu}$. We inject 32-dimensional Bernoulli noise to the fully connected layers. We learn the all model parameters for 2500 epochs with the learning rate 0.0005 and use the validation set for the early stopping. We use a two-stage learning process for SIG-VAE, SIVI-VGAE, and NF-VGAE. First, the embedding of each node is learned in the 128-dimensional latent space while injecting 5-dimensional Bernoulli noise to the system in the case of SIG-VAE and Naive SIG-VGAE. Then we use the learned embedding as node features for the second stage to learn 16 dimensional embedding while injecting more noise to SIG-VAE. We follow the same procedure for SIVI-VGAE too.

**Graph generation.** We have not specifically tuned the model but directly adopt the implementation setups for link prediction with and without node attributes.

**Node classification and graph clustering.** We use two GCN layers with sizes equal to [32, 16] followed by a fully connected layer with sizes 16 to infer $\boldsymbol{\mu}$. We inject 64-dimensional Bernoulli noise to the GCN layers. Learning rate is set to be 0.0005.

**Analysis of the complexity.** For the analysis of the real-world graph dataset Cora on a single GeForce GTX 1080 GPU node, it took 24.5, 11.7 , and 9.5 seconds for SIG-VAE, NF-VGAE, and VGAE methods with 100 epochs, respectively. For the analysis of the small real-world graph dataset NS on a same GPU node, it took 7.23, 7.84, and 7.09 seconds for SIG-VAE, NF-VGAE, and VGAE methods with 100 epochs, respectively.

# 7 Additional experimental results

## 7.1 Interpretable latent representations

In addition to the results of the **Swiss roll** graph in the paper, we also compare the latent representations of SIG-VAE and VGAE for a **torus** graph with 256 nodes connected by 512 edges as illustrated in Figure 1. We consider the coordinates of each node in $\mathbb{R}^3$ as node attributes for both methods in this experiment. We expect that the embedding of nodes to be symmetric since the graph itself is symmetric. We know that the inner-product decoder tries to embed a ring graph to a circle in space. Also, connected nodes should be in the same angle. Thus, the embedding of connected circles as in torus in $\mathbb{R}^3$ should be some lines coming out of center while their altitude is changing periodically. As we can see in Figure 1, SIG-VAE demonstrates a better latent representation than VGAE. To gain more insights about the posterior distributions, we show the distributions inferred by SIG-VAE and VGAE for three nodes in Figure 2. The inferred distributions are indeed skewed and multi-modal, very different from Gaussian. Being able to capture complex non-Gaussian distributions helps the model to represent the graph structure in a more meaningful way.

Figure 1: Torus graph (left) and its latent representation using SIG-VAE (middle) and VGAE (right). The latent representations (middle and right) are heat maps in $\mathbb{R}^3$. We expect that the embedding of the torus graph with the inner-product decoder to be multiple lines coming out of the center in $\mathbb{R}^3$, which is clearly better captured by SIG-VAE.

Figure 2: Latent representation distributions of three nodes in the torus graph using SIG-VAE (blue) and VGAE (red). SIG-VAE clearly infers more complex distributions that are multi-modal or skewed. This helps SIG-VAE to better represent the nodes in the latent space.

## 7.2 Link prediction

Table 2: AUC of link prediction in networks without node attributes. * indicates that the numbers are reported from Zhang and Chen [10].

| Data | MF* | SBM* | N2V* | LINE* | SC* | VGAE* | SEAL* | G2G | NF-VGAE | SIVI-VGAE | SIG-VAE(IP) | SIG-VAE |
|---|---|---|---|---|---|---|---|---|---|---|---|---|
| **USAir** | 94.08 | 94.85 | 91.44 | 81.47 | 74.22 | 89.28 | 97.09 | 92.17 | 95.74 | 94.22 | **97.56** | 94.52 |
| | ±0.80 | ±1.14 | ±1.78 | ±10.71 | ±3.11 | ±1.99 | ±0.70 | ±1.65 | ± 1.74 | ±0.43 | ±0.23 | ±0.28 |
| **NS** | 74.55 | 92.30 | 91.52 | 80.63 | 89.94 | 94.04 | 97.71 | 98.18 | 98.38 | 98.00 | **98.75** | **99.17** |
| | ±4.34 | ±2.26 | ±1.28 | ±1.90 | ±2.39 | ±1.64 | ±0.93 | ±0.51 | ±0.46 | ±0.34 | ±0.12 | ±0.45 |
| **Yeast** | 90.28 | 91.41 | 93.67 | 87.45 | 93.25 | 93.88 | 97.20 | 97.34 | 97.86 | 93.36 | **98.11** | 98.32 |
| | ±0.69 | ±0.60 | ±0.46 | ±3.33 | ±0.40 | ±0.21 | ±0.64 | ±0.32 | ±0.44 | ±0.63 | ±0.18 | ±0.26 |
| **Power** | 50.63 | 66.57 | 76.22 | 55.63 | 91.78 | 71.20 | 84.18 | 91.35 | 94.61 | 93.67 | 95.045 | 96.23 |
| | ±1.10 | ±2.05 | ±0.92 | ±1.47 | ±0.61 | ±1.65 | ±1.82 | ±0.41 | ±0.65 | ±0.78 | ±0.15 | ±0.12 |
| **Router** | 78.03 | 85.65 | 65.46 | 67.15 | 68.79 | 61.51 | 95.68 | 85.98 | 93.56 | 92.66 | **95.94** | **96.13** |
| | ±1.63 | ±1.93 | ±0.86 | ±2.10 | ±2.42 | ±1.22 | ±1.22 | ±1.25 | ±0.79 | ±0.25 | ±0.23 | ±0.26 |

Table 3: AP of link prediction in networks without node attributes. * indicates that the numbers are reported from Zhang and Chen [10].

| Data | MF* | SBM* | N2V* | LINE* | SC* | VGAE* | SEAL* | G2G | NF-VGAE | SIVI-VGAE | SIG-VAE(IP) | SIG-VAE |
|---|---|---|---|---|---|---|---|---|---|---|---|---|
| **USAir** | 94.36 | 95.08 | 89.71 | 79.70 | 78.07 | 89.27 | 95.70 | 90.22 | 96.27 | 94.48 | **97.50** | 94.95 |
| | ±0.79 | ±1.10 | ±2.97 | ±11.76 | ±2.92 | ±1.29 | ±2.61 | ±2.61 | ± 1.51 | ±0.80 | ±0.14 | ±0.28 |
| **NS** | 78.41 | 92.13 | 94.28 | 85.17 | 90.83 | 95.83 | 98.12 | 97.43 | 98.52 | 97.83 | **98.53** | **99.24** |
| | ±3.85 | ±2.36 | ±0.91 | ±1.65 | ±2.16 | ±1.04 | ±0.77 | ±2.34 | ±0.29 | ±0.40 | ±0.09 | ±0.40 |
| **Yeast** | 92.01 | 92.73 | 94.90 | 90.55 | 94.63 | 95.19 | 97.95 | 97.83 | 98.18 | 94.24 | **97.97** | **98.41** |
| | ±0.47 | ±0.44 | ±0.38 | ±2.39 | ±0.56 | ±0.36 | ±0.35 | ±0.28 | ±0.22 | ±0.46 | ±0.14 | ±0.13 |
| **Power** | 53.50 | 65.48 | 81.49 | 56.66 | 91.00 | 75.91 | 86.69 | 92.29 | 95.76 | 93.80 | **96.50** | **97.28** |
| | ±1.22 | ±1.85 | ±0.86 | ±1.43 | ±0.58 | ±1.56 | ±1.50 | ±0.37 | ±0.55 | ±0.83 | ±0.17 | ±0.30 |
| **Router** | 82.59 | 84.67 | 68.66 | 71.92 | 73.53 | 70.36 | 95.66 | 86.28 | 95.88 | 92.80 | **94.94** | **96.86** |
| | ±1.38 | ±1.89 | ±1.49 | ±1.53 | ±1.47 | ±0.85 | ±1.23 | ±1.32 | ±0.34 | ±0.18 | ±0.13 | ±0.27 |

More complete link prediction results with the standard deviation values from different runs are presented here. As we can see in Tables 2 and 3, SIG-VAE shows the consistent superior performance

Table 4: Graph clustering performance in citation networks with label.

| Method | Cora | | Citeseer | |
|---|---|---|---|---|
| | NMI | ACC | NMI | ACC |
| VGAE | 0.43 | 59.2 | 0.20 | 51.5 |
| **SIG-VAE** | **0.58** | **68.8** | **0.34** | **57.4** |

Table 5: Graph generation performance. The closest results to the original graph is highlighted in boldface.

| Detasets | Orignial Graph | | VGAE | | SIG-VAE (IP) | | SIG-VAE | |
|---|---|---|---|---|---|---|---|---|
| | Dens. | Clus. | Dens. | Clus. | Dens. | Clus. | Dens. | Clus. |
| **Cora** | 0.00143 | 0.24 | 0.1178 | 0.49 | 0.1178 | 0.49 | **0.00147** | **0.25** |
| **Citeseer** | 0.0008 | 0.14 | 0.09 | 0.45 | 0.26 | 0.42 | **0.0008** | **0.16** |
| **USAir** | 0.038 | 0.62 | 0.18 | 0.40 | 0.21 | **0.56** | **0.043** | 0.45 |
| **NS** | 0.002 | 0.63 | 0.36 | 0.47 | 0.26 | 0.42 | **0.02** | **0.49** |
| **Router** | 0.0004 | 0.01 | 0.16 | 0.49 | 0.16 | 0.49 | **0.0010** | **0.09** |

compared to the competing methods, especially over the baseline VGAE, in terms of both AUC and AP. It is interesting to note that, while the proposed sparse decoder works well for the sparser graphs, especially NS and Router sparse datasets, SIG-VAE with the inner-product decoder shows superior performance for the USAir graph which is much denser. Compared to the baseline VGAE, both SIVI-VGAE and NF-VGAE improve the results with a large margin in terms of both AUC and AP, showing the benefits of more flexible variational posterior. Comparing SIG-VAE with two other flexible inference methods shows that not only SIG-VAE is not restricted to the Gaussian assumption, which is not a good fit for link prediction with the inner-product decoder [2], but also it is able to model flexible posterior considering graph topology. The results for the link prediction of the Power graph clearly magnifies this fact as SIG-VAE improves the accuracy by 34% compared to VGAE.

## 7.3 Graph generation

In addition to the results of Cora dataset in the paper, we also used the inferred embedding representations of different graph dataset with and without node attributes to generate new graphs. Results are summarized in Table 5. The SIG-VAE results are much closer to the real-world graph in terms of both graph density and average clustering for very sparse graphs. For the USAir dataset, which is much dense compare to othe graphs, the average clustering coefficient of SIG-VAE with inner-product decoder is closer to the read-world graph. This can be describe the better link prediction results of SIG-VAE for USAir dataset. On the other hand, the generated graph by SIG-VAE with the Bernoulli-Poisson link decoder is much sparser as its density is very closer to the read-world graph. This shows the benefit of the proposed decoder to improve the flexibility of the generative model.

## 7.4 Graph clustering

SIG-VAE can be applied in the other application including graph clustering. We first tried SIG-VAE for getting low-dimential feature space and then apply Gaussian mixture clustering (GMM) on citation graphs with labels including Cora and Citeseer and compare its results with VGAE. We consider same number of parameters and GCN layer for both model. Results are summarized in Table 4. We report the normalized mutual information (NMI) and unsupervised clustering accuracy (ACC) of 10 runs. The decoders for both methods are inner-product decoder.

## 7.5 Drug-drug interaction network

Here, we also include the results on a drug-drug interaction network [8] capturing drug effect change due to the action of another drug. When several drugs are administered together, there might be adverse drug reactions due to drug-drug interactions. It is thus crucial to identify them during drug development. With a similar setup as in the paper, SIG-VAE achieves AUC and AP at 92.51 and 92.81, respectively. For comparison, VGAE gets 90.22 (AUC) and 90.29 (AP), respectively, and

GAE gets 90.73 (AUC) and 91.15 (AP). Hyperparameters are inherited from the original paper of each method.