[Reviews · NeurIPS 2019]

Reviewer 1



This paper proposes a Semi-Implicit VI extension of the GraphVAE model. SIVI assumes a prior distribution over the posterior parameter, enabling more flexible modeling of latent variables. In this paper, SIVI is straightforwardly incorporated into the Graph VAE framework. The formulation is simple but possibly new in the graph analysis literature. It is easy to understand the main idea. The proposed model shows good records in link prediction experiments. Fig. 3 is not reader-friendly in several aspects (i) the panels are simply too small. (ii) we can observe the posterior distributions learned by SIG-VAE is multi-modal. But the readers do not know that the posteriors of five nodes should be'' multi-modal. In other words, the SIG-VAE's variational posterior is closer to the true distribution, than that of VGAE? Are there any solutions that can answer this question more directly? I cannot fully understand the meaning of the result of graph generation experiments. What is the main message we can read from this result? I have a few questions concerning the result of the graph node classification experiment. (i) what kind of data splitting is employed in the experiments? (train/validation/test sample splitting) Data split has a huge impact on the final score. The split is the same with the standard split used in the Kipf-Welling's GCN paper? (ii) The performance of the proposed SIG-VAE is not so much impressive, compared to naive and simple GCN. Why is that? (iii) I think GCN is a strong baseline but not the best one to claim SOTA. In general, the [GAT] works better in many cases, including the cora and citeseer datasets. Please see the experiments in [Wu19]. [GAT] Velickovic+ Graph Attention Networks'', in Proc. ICML 2018 [Wu19] Wu+, Simpplifying Graph Convolution Networks'', in Proc. ICML 2019 + A combination fo Semi-implicit VI and graph VAE is new + Formulation is concise and easy to understand - Some unclear issues in Fig.3 and graph generation experiments, - The node classification result is not SOTA (too-strong claim) ### after author-feedback ### The authors provided satisfactory answers for some of my concerns. Considering the other reviewers' points of view at the same time, I raised the score.

Reviewer 2



Originality The paper is a combination of a number of ideas in the literature, where a careful combination of existing techniques leads to really good representation learning for graphs. In that sense the work is original and interesting. -----POST REBUTTAL----- I thank the authors for addressing my concerns / questions around a VAMP version of VGAE as well as questions around Eqn. 5. In general the rebuttal seems to include a lot of relevant experiments for the concerns from the review stage, and based on this evidence I am happy to keep my original score for the paper. Clarity The paper is generally clear and has clear mathematical formulations written down for all the methods considerered. Quality The paper has a number of thorough experiments and generally seems to be high quality in empirical evaluation. It also has a clear intuition for why the proposed method is better and extensively demonstrates and validates it. Significance The paper seems like a significant contribution to the graph representation learning literature. Weaknesses - It would be good to better justify and understand the bernoulli poisson link. Why are the number of layers used in the link in the poisson part? The motivation for the original paper [40] seems to be that one can capture communities and the sum in the exponential is over r_k coefficientst where each coefficient corresponds to a community. In this case the sum is over layers. How do the intuitions from that work transfer here? In what way do the communities correspond to layers in the encoder? It would be nice to beter understand this. Missing Baselines - It would be instructive to vary the number of layers of processing for the representation during inference and analyze how that affects the representations and performance on downstream tasks. - Can we run VGAE with a vamp prior to more accurately match the doubly stochastic construction in this work? That would help inform if the benefits are coming from a better generative model or better inference due to doubly-semi implicit variational inference. Minor Points - Figure 3: It might be nice to keep the generative model fixed and then optimize only the inference part of the model, parameterizing it as either SIG-VAE or VGAE to compare the representations. Its impossible to know / compare representations when the underlying generative models are also potentially different.

Reviewer 3



The paper is incremental work compared to the Semi-Implicit VAE [38]. The general idea of the SIVAE is to model the parameters of the VAE model ($\psi$) as a random variable that one can sample from but does not necessarily have an explicit form which results into a mixture like behavior for the VAE. In this work, the authors propose to use that framework for Graph data. The idea is to sample from $\psi$ and concatenate with each layer of graph VAE. The rest follows [38]. They also propose another variant based on the normalized flow which read a bit out of sync (afterthought/add-on) with the rest of the paper.

Reviewer 4



### edit after author response ### I read the feedback and found the additional results impressive. I am still uncertain about \psi: l116 says "implicit prior distribution parametrized by \psi" (I assumed this means q(Z|\psi)) and "reparametrizable q_\phi(\psi|X,A)" and the feedback states \psi is \mu and \Sigma. I think Gaussian random variable Z is reparametrizable and q(Z|\psi) is explicit. Since the major strength of this paper is the empirical performance, the clarity of method/model description and experimental setups (mentioned by other reviewers) are very important. I hope the reviews are helpful for improving the presentation. ########################### This paper presents SIG-VAE for learning representations of nodes in a given graph. This method is an extension of VGAE in two ways. The use of semi-implicit variational distribution enriches the complexity of variational distribution produced by the encoder. The decoder uses the Bernoulli--Poisson likelihood to better fit the sparse link structure. Experiments in Section 4 compares SIG-VAE with many modern methods on various tasks and datasets. My questions and suggestions are as follows. * The reviewer failed to comprehend the connection between the semi-implicit distribution in Eq.(2) and the following part lines 122--142. Specifically, an absense of \psi in Eqs. (3,4) is confusing. Is \epsilon_u ~ q_u(\epsilon) interpreted as \psi ~ q_\phi of Eq. (2)? If so, \psi conditioned on X, A is misleading information. * Figure 1 is not informative either. The caption says 'diffuses the distributions of the neighboring nodes', but VGAE already achieves it: the latent representaiton is propagated to adjacent nodes, and it infers distributions. What does SIG-VAE attain on top of it? The illustration is also obscure. Does this tell that the distribution of latent representaiton of each node can be multimodal and that the neighboring distributions influence the distribution of certain nodes? * CONCAT operators are unclear in Eqs. (3,4). In particular, I want to know the size of \epsilon_u and h_{u-1} to see what information is added to X. After checking the code, I assumed \epsilon_u is concatenated to node feature for each node and layer, but not convinced due to unfamiliarity with TensorFlow. Why is X fed in every layer? For u>1, h_u seems to carry some information on X. * Dots in middle and right panels of Figure 2. Suppopsed the latent representation is infered as distributions, what are the dots in the figure? Mean values? As visualized in Figure 3, the distributions may take multiple modes. I am curious if it is okay to shrink such distributions to a single point, though the full visualization is challenging. * Evaluation on graph generation task. Density and average clustering coefficient are adopted as metrics of the graph generation task in Section 4.3. While these scores indicate the efficacy of Bernoulli--Poisson likelihood for sparse edges, they may not fully characterize graph structures. Using the following metrics may further strengthen the result: MMD-based score in Section 4.3 of [a], and KL-divergence between the degree distributions [b]. [a] J. You et al. GraphRNN: Generating Realistic Graphs with Deep Auto-regressive Models, ICML 2018. [b] Y. Li et al. Learning Deep Generative Models of Graphs https://arxiv.org/abs/1803.03324

[Author Response · NeurIPS 2019]

We appreciate all four reviewers' comments. Due to space limit, we focus on addressing major ones. We'd like to reiterate the novelty and significance of our key contribution: SIG-VAE integrates a carefully designed generative model, well suited to real-world sparse graphs, and a sophisticated variational inference network, which propagates the graph structural information and distribution uncertainty to capture complex posterior. SIG-VAE clearly outperforms a simple combination of SIVI (or NF) and VGAE that does not propagate uncertainty in its inference network, and provides much more interpretable latent representations than VGAE. As a flexible generative model, SIG-VAE outperforms SOTA methods in link prediction by a large margin. In addition, it is comparable with SOTA when modified to perform two additional tasks (node classification and graph clustering), even though these two tasks are more suited to supervised learning methods.

Regarding Fig.3 and interpretability (**R1,R4**), we have run HMC to infer posteriors (not shown here due to space limit but will be added into revision), confirming that the SIG-VAE's variational posterior is closer to the HMC inferred posterior, in particular in capturing multi-modality, skewness, and sharp and steep changes. To explain why multi-modality may arise, we used Asynchronous Fluid [Parés et al., 2017] to visualize the Swiss Roll graph by highlighting detected communities with different colors. The three red (two orange) nodes are the nodes with multi-modal (skewed) distributions in Fig. 3 of the paper. These nodes with multi-modal posteriors reside between different communities.

For graph generation (**R1,R5**), comparing the statistics of the generated and training graphs is a standard way for model checking. A well-trained good generative model such as SIG-VAE will have characteristics of generated graphs resemble these of the true one , and paves a way for new applications such as discovering new drugs. Following the instruction of **R5**, we compare KL(node degree distribution of generated graph ‖ that of true). The {SIG-VAE, SIG-VAE (IP)} results are {3.7e-07, 0.33} and {1.4e-06, 0.60} for Cora and Citeseer, respectively, clearly showing the advantage of SIG-VAE using the Bernoulli-Poisson link decoder. We will also add the MMD scores into the revision.

Data splitting (**R1**) is the same as GCN [Kipf & Welling, 2017]. Regarding node classification performance (**R1**), as stated in L301, GCN is a (semi-)supervised model for node classification while ours is a generative model. We will revise L310 to clarify *outperforming* SOTA refers to link prediction. **R1** asked to include two additional baselines. The accuracy (%) of GAT is 83.0, 72.5 & 79.0 for Cora, Citeseer & Pubmed, respectively (note GAT uses 64 hidden features, while the other methods including SIG-VAE use 16). SGC [Wu et al., 2019] gets 81.0, 71.9 & 78.9. SIG-VAE's results are 79.7, 70.4 & 79.3, close to SOTA, despite not being trained for this task in a supervised way.

We provide clarifications for Sec. 3 (**R5**): 1) $\psi$ in eq 2 consists of $\boldsymbol{\mu}$ and $\boldsymbol{\Sigma}$ in eqs 3-4. 2)-3) In SIG-VAE, we mix and propagate the representational uncertainty across the graph while in VGAE only deterministic features are mixed. We showed that propagating distributions across graph is beneficial by comparing SIG-VAE with Naive-SIG-VAE and NF-VGAE. Fig. 1 illustrates that neighboring distributions influence the distribution of certain nodes in SIG-VAE. We note that this is not the case for Naive-SIG-VAE, NF-VGAE and VGAE where deterministic features are propagated. 4)-5) The dimension of $\boldsymbol{\epsilon}_u$ is $N \times d^{(n)}$ where $d^{(n)}$ (noise dimension) is a hyperparameter. We add noise to each layer as a part of semi-implicit construction. The dimension of $\mathbf{h}_{u-1}$ is $N \times d_{u-1}^{(h)}$ where $d_{u-1}^{(h)}$ is the number of graph convolutional filters at hidden layer $u - 1$. While in eqs 3-4, we used skip connection (concatenation with $\mathbf{X}$), in our experiments (submitted code) we didn't use skip connection since we only used 2 layers for a fair comparison with the baselines. Our experiments for deeper structures of SIG-VAE showed skip connection improves the performance.

GCN-AE (GAE) is the link prediction version of GCN. While we already reported the results of GAE for Table 1 in our submission, we will include the results for Table 2, as **R4** instructed. The mean of AUCs for 10 runs are 93.09, 93.14, 93.74, 72.21 & 55.73 for USAir, NS, Yeast, Power & Router datasets, respectively. The AP results are 95.14, 95.26, 95.34, 77.13 & 67.50. **R4** also asked for a real application. We here include the results on a drug-drug interaction network capturing drug effect change due to the action of another drug. When several drugs are administered together, there might be adverse drug reactions due to drug-drug interactions. It is thus crucial to identify them during drug development. With a similar setup as in the paper, SIG-VAE achieves AUC and AP at 92.51 and 92.81, respectively. For comparison, VGAE gets 90.22 (AUC) and 90.29 (AP), respectively, and GAE gets 90.73 (AUC) and 91.15 (AP). Hyperparameters are inherited from the original paper of each method (**R4**), we will add details in the supplement.

It appears that our notation (using $l$ for latent dimension of $\mathbf{Z}$ and $L$ for number of layers) led to confusion on eq 5 (**R2**). Different from Zhou [2015] that decomposes the Poisson rate in an additive way, here we decompose it in a multiplicative way (additive inside the exponential), which removes the non-negative constraint on $\boldsymbol{z}_i$ and no longer provides the same community structure interpretation as in Zhou [2015]. The AUC results for Power dataset are 94.34, 96.23 & 96.37 for 8-, 16- & 32-dimensional latent space. The AP results are 94.70, 97.28 & 97.42, respectively.

**R2** suggested trying VGAE + VAMP prior, i.e., replacing $p(z) = N(0,1)$ in VGAE with $p(z) = \sum_k q_\phi(z|u_k)$ , where $(u_1,...,u_K,\phi)$ will be treated as variational parameters to be optimized. The non-trivial part is how to define $u_k$. In the VAMP prior paper, $u_k$ will be the same dimension as an input data $x_i$, but here the inputs are $X$ and $A$. On the other hand, if VAMP prior helps VGAE, then (semi-implicit) VAMP prior is likely to also help SIG-VAE (semi-implicit VAMP prior can be inferred with doubly SIVI). These potential extensions will be discussed and suggested for future study.

[Meta-Review · NeurIPS 2019]

The paper "carefully designed the SIG-VAE model" and showed SOTA results. We judged that the contributions are significant, but reviewers raised the following concerns. - No technical novelty in SIVI+GVAE (= naive SIG-VAE in Section 3.2). - Contributions are (non-naive!) SIG-VAE modeling (3-4) for graph analysis, giving strong empirical result (SOTA in link prediction, comparable in node classification, etc.) - Contribution in interpretability is not convincing. We strongly recommend the authors to revise the paper significantly. The biggest problem is in organization. The paper is written as if the main contribution is SIVI+GVAE, which is straightforward as reviewers pointed out. The main contribution (what the authors say in the rebuttal "careful design of SIG-VAE") must be appropriately highlighted. - Section 3.1 should be shrunken and moved to Section 2, since nothing there is novel. - Contributions would become clearer if the authors set the naive methods in Section 3.2 as baselines. - Highlight not the methodology (SIVI+GVAE) but the modeling (3-4). The second sentence in the rebuttal "SIG-VAE integrates a carefully designed generative model" really explains the main contribution. In the submitted version, the "carefully designed" modeling is not carefully explained. The authors say propagating uncertainty is essential. But is (3-4) only the way to propagate uncertainty? Or details don't matter if uncertainty is propagated? This is the most important point. If there are many possibilities and the authors made a particular choice, they should justify it. Also in Line 257-261 the authors explain their two-stage learning for the case without node attributes, without any justification.